# Highly Efficient Liquid-Phase Hydrogenation of Naringin Using a Recyclable Pd/C Catalyst

**DOI:** 10.3390/ma12010046

**Published:** 2018-12-24

**Authors:** Jiamin Zhao, Ying Yuan, Xiuhong Meng, Linhai Duan, Rujin Zhou

**Affiliations:** Department of Chemical Engineering, Guangdong University of Petrochemical Technology, Maoming 525000, China; zhaojiamin08031223@163.com (J.Z.); yuanyingsdlc@163.com (Y.Y.); mengxiuh@163.com (X.M.)

**Keywords:** Pd/C, Catalyst, Hydrogenation, Naringin, Dihydrochalcone

## Abstract

A highly efficient liquid-phase hydrogenation reaction using a recyclable palladium on carbon (Pd/C) catalyst has been used for the transformation of naringin to its corresponding dihydrochalcone. The effects of various solvents on the hydrogenation process were studied, with water being identified as the optimal solvent. The analysis also revealed that sodium hydroxide (NaOH) can accumulate on the surface of the Pd/C catalyst in alcoholic solvents, leading to its inactivation. The higher solubility of NaOH in water implies that it remains in solution and does not accumulate on the Pd/C catalyst surface, ensuring the catalytic activity and stability.

## 1. Introduction

There has been significant interest in dihydrochalcones (DHC) since Horwits reported their strong sweetness in 1963 [1]; these heterocycles are also known to exhibit antioxidant or antidiabetic activities [2,3]. For example, Chao et al. [4] used phlorizin as a DHC template to develop three antidiabetic drugs approved by the Food and Drug Administration (FDA) and the European Medicine Agency (EMA) [5]. Snijman et al. [6] reported that the dihydrochalcones aspalathin and nothofagin exhibit strong antioxidant activities, implying that they are ingested into the human body as intact glycosides. Janvier et al. [7] gained approval for using the dihydrochalcone neohesperidin as a food additive (E959) in Europe. Naringin is a flavanone occurring as the major component of exocarpium and pomelo, which are cultivated widely in south China [8,9,10]. Therefore, processes to convert naringin to value-added products on an industrial scale must be developed.

Naringin dihydrochalcone (NDC) can be prepared from naringin via a two-step alkali- mediated hydrogenation reaction at high pressures [11]. First, NaOH(aq) or (KOH(aq)) is used as a base for ring-opening of the tetrahydropyran-4-one fragment of naringin to afford the corresponding chalcone intermediate with an (E)-alkene bond. Second, selective hydrogenation of this alkene bond using Pd/C (or Raney nickel) catalyst at high hydrogen pressures yield the dihydrochalcone product; the reaction scheme is shown in Scheme 1. The key feature of this reaction is the selective hydrogenation of the C=C bond of the DHC intermediate over its C=O functionality. An ideal process would involve hydrogenation at medium-to-low hydrogen pressures at an ambient temperature using a heterogeneous catalyst that can be easily recovered and recycled [12,13]. Pd/C has often been used as a catalyst to selectively reduce the C=C bond of α, β-unsaturated carbonyl compounds, and the efficacy and mechanism of this reaction has been proven by experimental and theoretical results [14,15,16] and studies [17,18,19].

The choice of an appropriate solvent is critical for the success of hydrogenation using heterogeneous catalysis; the physical properties of solvents can significantly influence the yield and distribution of the hydrogenated products. An ideal solvent in a catalytic reaction can dissolve the solid reactants and products effectively, enhance conversion rates, and ensure that the catalyst surface remains free of inhibitors [20]. Alcohols (e.g., methanol or ethanol) are the most widely used solvents for reported heterogeneous catalytic hydrogenation reactions [21]. Wei et al. [22] have shown that Pd catalysts exhibit higher activity for hydrogenation reactions in toluene and ethanol, than in DMF, acetonitrile, or water. However, Ma et al. [23,24] reported that water was a more effective solvent for liquid-phase hydrodechlorination of chlorophenols catalyzed by Raney nickel. However, there has been little research on the effects of different solvents on liquid-phase catalytic hydrogenation of naringin.

In this study, our primary aim was to develop a practical process for catalytic hydrogenation of naringin to afford naringin dihydrochalcone using Pd/C as a recyclable heterogeneous catalyst. This was achieved by studying the effects of different solvents and the influence of water content on the yield of hydrogenation. The effects of solvents and reaction conditions on the structure of the recovered Pd/C catalyst was also studied using transmission electron microscopy (TEM), X-ray photoelectron spectroscopy (XPS), etc.

## 2. Experimental Section

### 2.1. Materials and Methods

The 732 cation exchange resin was purchased from Shape Chemical, Shanghai, China. Naringin was obtained from the National Institute for the Control of Pharmaceutical and Biological Products in Beijing, China. Other chemicals of analytical grade were purchased from Sigma-Aldrich, St. Louis, MO, USA.

TEM characterization was performed using a Hitachi HT-7700 microscope (Hitachi Corporation, Tokyo, Japan). High-resolution TEM (HRTEM) was performed using a Tecnai G2 F30 S-Twin microscope (Thermo Fisher Scientific, MA, USA) operated at an acceleration voltage of 300 kV.

The surface composition of the catalyst was analyzed using XPS (Thermo Escalab 250Xi XPS) (Thermo Fisher Scientific, MA, USA) with Al Kα radiation, operated at 15 kV and 14.9 mA, as the excitation source. Binding energy values were referenced to the C (1s) peak at 285.0 eV.

### 2.2. Preparation of Pd/C Catalyst

The 10 wt % Pd/C catalyst was prepared by adding 0.2 g of activated carbon to 40 mL of water containing 0.11 g Na_2_PdCl_4_ (Sigma-Aldrich, St. Louis, MO, USA). After sonication for 10 min and stirring for 4 h, NH_3_·H_2_O (Sigma-Aldrich, St. Louis, MO, USA) was added dropwise to the solution for adjusting pH to 8.5. Next, the Pd adsorbed on activated carbon was reduced by adding 10 mL of freshly prepared 0.08 mol L-1 NaBH_4_ (Sigma-Aldrich, St. Louis, MO, USA) solution. The mixture was further stirred for 12 h. Finally, the Pd/C was washed and collected by several rounds of centrifugation and the final product was dried under vacuum for 12 h.

### 2.3. Procedures Used to Carry Out Hydrogenation Reactions

Liquid-phase hydrogenation was performed in a 250-mL Hastelloy autoclave (Parr, Illinois, USA) fitted with a mechanical stirrer and an electric temperature controller. In a typical reaction, the reactor was loaded with 0.50 g of the reduced heterogeneous Pd/C catalyst and 150 mL of an aqueous solution containing 5.0 g of naringin and 0.6 g of NaOH (Sigma-Aldrich, St. Louis, MO, USA).

The reactor was purged with H_2_ to remove air and was then pressurized to 1.5 MPa using H_2_. The reactor was then heated to 40 °C and stirred at 600 rpm for 10–14 h. The reaction mixture was then filtered to recover the heterogeneous catalyst and the filtrate was passed through a type 732 cation exchanger (Shape Chemical, Shanghai, China) (to exchange Na^+^ for H^+^ cations). This resulted in crystallization of the dihydrochalcone product from the solution, and the crystals were collected using vacuum filtration [8].

## 3. Results and Discussion

### 3.1. Characterization of Pd/C and the Product DHC

The TEM images (Figure 1) of the as-synthesized Pd/C indicated that Pd nanoparticles with diameters of 4.3 nm are dispersed uniformly on the carbon black. The X-ray diffraction (XRD) patterns of the Pd/C sample (Figure 2a) exhibit three typical peaks at 2θ = 40.1°, 47.4°, and 68.7°, corresponding to the (111), (200), and (220) reflections, respectively, of crystalline Pd, which matches well with PDF 65-6174. The diameters of the Pd nanoparticles can also be measured by XRD using the Scherrer equation [25]: L=Kλ/βcosθ. Where λ is the X-ray wavelength in nanometer (nm), β is the peak width of the diffraction peak profile at half maximum height resulting from small crystallite size in radians and K is a constant related to crystallite shape, normally taken as 0.89. The diameter of the Pd nanoparticle calculated using Scherrer equation is 0.41 nm, which is similar to the average particle size (0.43 nm) observed from TEM. The XPS spectrum of Pd/C (Figure 2b) shows two symmetrical peaks assigned to the Pd 3d 5/2 and Pd 3d 3/2 core levels. The peaks at 335.9 and 341.4 eV are attributed to metallic Pd0, while those at 337.3 eV and 343.4 eV correspond to the Pd 2+ species. The percentage of metallic Pd0 was calculated from the relative areas under these peaks; metallic Pd0 was found to be the main metal species on the surface of the as-prepared catalyst (65 wt %).

The product naringin hydrochalcone was characterized by infrared spectroscopy (IR) absorption (Figure 3). IR absorption at 3390 cm^−1^ (-OH), 2923.84 cm^−1^ (-CH_3_), 1631 cm^−1^ (conjugated -C=O), 1513 cm^−1^, 1438 cm^−1^ and 817 cm^−1^ (aromatic nucleus) were indicative of a hydroxylated dihydrochalcone, similar to the result of Tang et al. [8].

### 3.2. Solvent Effect on the Hydrogenation Reaction of Naringin over Pd/C Catalyst

The hydrogenation of naringin over Pd/C catalyst was examined in various solvents, including methanol, ethanol, iso-propanol, and n-hexane (Figure 4a). It was found that naringin could be hydrogenated in all these solvents, with DHC yields depending strongly on the solvent, according to the following order: methanol > ethanol > iso-propanol > n-hexane > toluene. The Pd/C catalyst showed greater activity in alcoholic solvents than in n-hexane and toluene. Therefore, protic solvents are better solvents than aprotic ones for liquid-phase hydrogenation reactions of naringin. These results are consistent with previous reports, which describe that hydrogenation of non-polar substrates in polar solvents give the best yields of hydrogenated products [20,26]. The rates of these hydrogenation reactions were also dependent on solvent polarity, with hydrogenation reactions occurring faster in higher-polarity solvents. Indeed, for alcoholic solvents, DHC yield was directly proportional to the order of the normalized empirical parameter (ETN), dielectric constant (ε), and dipole moment (μ) of the solvents [23,27,28].

Water, a highly polar protic solvent, can be used as an additive to increase the polarity of alcoholic solvents. Therefore, a series of mixed alcohol/water solvents were investigated to determine the impact of solvent polarity on hydrogenation reactions. A series of Pd/C-catalyzed hydrogenation reactions of naringin were performed in different mixed solvents, i.e., ethanol, 20 wt % water-ethanol (20/80, *v*/*v*), 50 wt % water-ethanol (50/50, *v*/*v*), 80 wt % water-ethanol (80/20, *v*/*v*), and water (Figure 4b). These experiments revealed that the highest DHC yield was obtained using water as the solvent, followed by progressively lower DHC yields for 80 wt % water-ethanol, 50 wt % water-ethanol, 20 wt % water–ethanol, and ethanol. These results showed that the rate of hydrogenation increased with increasing polarity of the mixed ethanol-water solvents. Therefore, the rate of hydrogenation of naringin over the Pd/C catalyst can be increased significantly by simply adding water to the alcoholic solvent.

The stability and recyclability of the 10 wt % Pd/C catalyst used for hydrogenation in water, 50 wt % water-ethanol (50/50, *v*/*v*), and ethanol were then investigated (Figure 5). In ethanol, the initial DHC yield was 70 wt % after 12 h; however, this yield dropped to 59 wt % on using Pd/C catalyst that had been recycled five times. However, the DHC yield obtained using water or water-ethanol (50/50, *v*/*v*) as solvent remained almost unchanged after six rounds of catalyst recycling, indicating superior stability of the Pd/C catalyst in these solvents. Therefore, we concluded that water was the best solvent for performing Pd/C-catalyzed liquid-phase hydrogenation of naringin.

### 3.3. The Influence of Solvent on Pd/C Catalyst Structure

Having shown that among the solvents studied, water is the optimal solvent for hydrogenation of naringin [20,28,29,30], we used TEM and XPS studies to elucidate the solvent effect on the structure of the Pd/C catalyst. The recovered 10 wt % Pd/C catalyst after six rounds of use for hydrogenation of naringin was analyzed using TEM. The surface morphologies of 10 wt % Pd/C catalyst that had been reused six times using water (Figure 6b) and 50 wt % water-ethanol (Figure 6c) as solvent were almost identical to that of the original Pd/C catalyst. On the other hand, when ethanol was used as the solvent, the morphology of the recovered Pd/C catalyst changed significantly, with significant amounts of crystalline material deposited on its surface after six rounds of use in the hydrogenation reactions (Figure 6d).

To gain more information about the structural composition of this crystalline material, 10 wt % Pd/C catalysts recovered from ethanol–water (50/50) and ethanol reactions were subjected to XPS analysis. Table 1 provides the XPS analysis results of the composition of fresh and reused Pd/C catalysts. This table shows the presence of Pd3d and O1s emissions at 336.39 eV and 532.60 eV, respectively, probably due to the presence of PdO. The biggest difference between catalyst samples recovered from hydrogenation reactions in water and ethanol was in their Pd3d and Na1s emissions. The Pd-to-Na ratios of a Pd/C catalyst recovered from hydrogenation reactions in water and ethanol were 1.79:1 and 0.38:1, respectively. This indicates that the crystalline material observed on the surface of the catalyst recovered from hydrogenation reactions in ethanol is probably NaOH. Therefore, the reduced catalytic activity of Pd/C in ethanol is due to NaOH blocking the catalytically active sites on the catalyst surface, preventing it from effectively absorbing hydrogen and the naringin substrate. However, when water is present, its higher polarity ensures that NaOH remains in solution, preventing the Pd/C catalyst surface from becoming deactivated.

### 3.4. Optimal Reaction Conditions for Naringin Hydrogenation

The effects of reaction temperature, pressure, pH, and catalyst content on the DHC yield in hydrogenation of naringin were then investigated (Figure 7). Figure 7a shows that the DHC yield in these reactions increased up to a maximum at hydrogen pressure of 2.0 MPa, with no further increase in yield at higher pressures. Figure 7b shows that the optimal temperature for these hydrogenation reactions was 42–45 °C. Higher temperatures resulted in catalyst degradation, leading to lower DHC yields. Increasing the amount of catalyst from 2 wt % to 10 wt % significantly increased the DHC yield, while the optimum basicity of the NaOH(aq)solvent was found to be pH 11.8 (Figure 7d). The role of NaOH on the reaction is to open heterocycles of naringin for forming the corresponding chalcone intermediate. If the pH is lower than 11, the heterocycles cannot be opened. If the pH is too high, the glycoside bonds on naringin may be destroyed, and the corresponding chalcone intermediate cannot be formed. These results were combined to design the optimal conditions for this hydrogenation reaction: hydrogen reaction pressure of 2.0–2.5 MPa, reaction temperature of 40–45 °C, pH 11.5–12.0, and Pd/C catalyst loading of 8–10 mol wt %.

## 4. Conclusions

The effects of various solvent systems on the Pd/C-catalyzed liquid-phase hydrogenation of naringin have been investigated. Water was found to be the optimal solvent for affording the corresponding dihydrochalcone in good yield. The presence of water was critical for preventing accumulation of NaOH on the surface of the Pd/C catalyst, which can lead to catalyst deactivation in alcoholic solvents. An optimal high-yielding and scalable process was developed using a hydrogen pressure of 2.0–2.5 MPa, a reaction temperature of 40–45 °C, an aqueous solvent of pH 11.5–12.0, and a recyclable Pd/C catalyst loading of 8–10 wt %.

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
