# Peer review of "Highly Efficient Liquid-Phase Hydrogenation of Naringin Using a Recyclable Pd/C Catalyst"

_materials, 2018, doi:10.3390/ma12010046_

Round 1
Reviewer 1 Report
This manuscript reports the synthesis of naringin hydrochalcone by one-pot hydrolysis-hydrogenation of naringin with Pd/C+NaOH catalyst and H2 under mild conditions. The experimental work seems correct and the manuscript is easy-to-follow. The results are of interest, since they allow the synthesis of this artificial sweetener in just one step instead of the current two-step synthesis. However, some issues should be addressed before publication:
- Characterization: Does the value of the Scherrer equation from the XRD match the experimental average particle size observed? Peaks in the XRD are quite broad, which usually indicate smaller particles that could not be seen within the TEM resolution. What about commercially available Pd/C? It should be similarly active as a catalyst.
- The reaction scheme should be included: Narginin to hydrochalcone, as well as the role of Pd/C-H2 and base during reaction, highligthening the selectivity of the reaction.
- Does the pH have any influence not only in the deprotonation of the alpha-carbon to the ketone but also in the phenol functionalities? What is the exact nature of narginin at this pH, is it deprotonated?
- Characterization of the product naringin hydrochalcone should be given
- Some typos: Section 2.3 “heterogeneous catalyst”; page 4 “Table 1”
Author Response
Those comments are all valuable and very helpful for revising and improving our paper, as well as the important guiding significance to our researches. We have studied comments carefully and have made correction which we hope meet with approval. Revised portion are marked in red in the paper. The main corrections in the paper and the responds to the reviewer’s comments is in the attached file.

Reviewer 2 Report
Very nice and very complete paper with extensive characterisation and very interesting results. I would just sugges to add an illustration dealing with the structure of naringin and the reaction scheme to better understand the reaction.
Author Response
Those comments are all valuable and very helpful for revising and improving our paper, as well as the important guiding significance to our researches. We have studied comments carefully and have made correction which we hope meet with approval. Revised portion are marked in red in the paper. The main corrections in the paper and the responds to the reviewer’s comments are in the attached file.
